# INVESTIGATING THE FAIRNESS OF LARGE LANGUAGE MODELS FOR PREDICTIONS ON TABULAR DATA

## ABSTRACT

Recent literature has suggested the potential of using large language models (LLMs) to make predictions for tabular tasks. However, LLMs have been shown to exhibit harmful social biases that reflect the stereotypes and inequalities present in the society. To this end, as well as the widespread use of tabular data in many high-stake applications, it is imperative to explore the following questions: what sources of information do LLMs draw upon when making predictions for tabular tasks; whether and to what extent are LLM predictions for tabular tasks influenced by social biases and stereotypes; and what are the consequential implications for fairness? Through a series of experiments, we delve into these questions and show that LLMs tend to inherit social biases from their training data which significantly impact their fairness in tabular prediction tasks. Furthermore, our investigations show that in the context of bias mitigation, though in-context learning and fine-tuning have a moderate effect, the fairness metric gap between different subgroups is still larger than that in traditional machine learning models, such as Random Forest and shallow Neural Networks. This observation emphasizes that the social biases are inherent within the LLMs themselves and inherited from their pre-training corpus, not only from the downstream task datasets. Besides, we demonstrate that label-flipping of in-context examples can significantly reduce biases, further highlighting the presence of inherent bias within LLMs.

## 1 INTRODUCTION

Many recent works propose to use large language models (LLMs) for tabular prediction (Slack & Singh, 2023; Hegselmann et al., 2023), where the tabular data is serialized as natural language and provided to LLMs with a short description of the task to solicit predictions. Despite the comprehensive examination of fairness considerations within conventional machine learning approaches applied to tabular tasks (Bellamy et al., 2018), the exploration of fairness-related issues in the context of employing LLMs for tabular predictions remains a relatively underexplored domain.

Previous research has shown that LLMs, such as GPT-3 (Brown et al., 2020), GPT-3.5, GPT-4 (OpenAI, 2023) can exhibit harmful social biases (Abid et al., 2021a; Basta et al., 2019), which may even worsen as the models become larger in size (Askell et al., 2021; Ganguli et al., 2022). These biases are a result of the models being trained on text generated by humans that presumably includes many examples of humans exhibiting harmful stereotypes and discrimination and reflects the biases and inequalities present in society (Bolukbasi et al., 2016; Zhao et al., 2017), which can lead to perpetuation of discrimination and stereotype (Abid et al., 2021a; Bender et al., 2021).

Considering that tabular data finds extensive use in high-stakes domains (Grinsztajn et al., 2022) where information is typically structured in tabular formats as a natural byproduct of relational databases (Borisov et al., 2022), it is of paramount importance to thoroughly examine the fairness implications of utilizing LLMs for predictions on tabular data. In this paper, we conduct a series of investigation centered around this critical aspect, with the goal of discerning the underlying information sources upon which LLMs rely when making tabular predictions. Through this exploration, our investigation aims to ascertain whether, and to what degree, LLMs are susceptible to being influenced by social biases and stereotypes in the context of tabular data predictions.

Through experiments using GPT-3.5 to make predictions for tabular data in a zero-shot setting, we demonstrate that LLMs exhibit significant social biases (Section 4). This evidence confirms that

LLMs inherit social biases from their training corpus and tend to rely on these biases when making predictions for tabular data.

Furthermore, we demonstrate that providing LLMs with few-shot examples (in-context learning) or fine-tuning them on the entire training dataset both exhibit moderate effect on bias mitigation (Sections 5.1 and 6.1). Nevertheless, the achieved fairness levels remain below what is typically attained with traditional machine learning methods, including Random Forests and shallow Neural Networks, once again underscoring the presence of inherent bias in LLMs. Additionally, our investigation further reveals that flipping the labels of the in-context examples significantly narrows the gap in fairness metrics across different subgroups, but comes at the expected cost of a reduction in predictive performance. This finding, in turn, further emphasizes and reaffirms the indication of inherent bias present in LLMs (Section 5.2). Additionally, we further show that while resampling the training set is a known and effective method for reducing biases in traditional machine learning methods like Random Forests and shallow Neural Networks, it proves to be less effective when applied to LLMs (Section 6.2).

These collective findings underscore the significant influence of social biases on LLMs' performance in tabular predictions. These biases significantly undermines the fairness and poses substantial potential risks for using LLMs on tabular data, especially considering that tabular data is extensively used in high-stakes domains, highlighting the need for more advanced and tailored strategies to address these biases effectively. Straightforward methods like in-context learning and data resampling may not be sufficient in this context.

## 2 RELATED WORK

### 2.1 FAIRNESS AND SOCIAL BIASES IN LLMS

Fairness is highly desirable for ensuring the credibility and trustworthiness of algorithms. It has been demonstrated that unfair algorithms can reflect societal biases in their decision-making processes (Bender et al., 2021; Bommasani, 2021), primarily stemming from the biases present in their training data (Caliskan et al., 2017; Zhao et al., 2017). LLMs, pre-trained on vast natural language datasets, are particularly susceptible to inheriting these social biases and have been shown to exhibit biases related to gender (Lucy & Bamman, 2021), religion (Abid et al., 2021b) and language variants (Ziems et al., 2023; Liu et al., 2023a). These social biases can lead to perpetuation of discrimination and stereotype (Abid et al., 2021a; Bender et al., 2021; Weidinger et al., 2021). While recent literature has made strides in addressing these issues, there still exists a significant gap in comprehensively assessing fairness in LLMs and its mitigation strategies for tabular data.

### 2.2 TABULAR TASKS AND LLM FOR TABULAR DATA

Tabular data extensively exist in many domains (Shwartz-Ziv & Armon, 2021). Previous works propose to utilize self-supervised deep techniques for tabular tasks (Yin et al., 2020; Arik & Pfister, 2021), which, however, still underperform ensembles of gradient boosted trees in the fully supervised setting (Grinsztajn et al., 2022). This disparity in performance can be attributed to the locality, sparsity and mixed data types of tabular data. In recent times, LLMs have undergone intensive training using vast amounts of natural language data, which has enabled them to exhibit impressive performance across various downstream tasks (Brown et al., 2020; OpenAI, 2023), even with little or no labeled task data. Therefore, recent approaches by Hegselmann et al. (2023); Slack & Singh (2023) suggests serializing the tabular data as natural language, which is provided to LLM along with a short task description to generate predictions for tabular tasks.

However, tabular data plays a crucial role in numerous safety-critical and high-stakes domains (Borisov et al., 2022; Grinsztajn et al., 2022), which makes the fairness particularly crucial when employing LLMs for making predictions on tabular data, especially considering the inherent social biases present in LLMs. Despite the importance, this still remains largely unexplored. To the best of our knowledge, we regard our work as one of the most comprehensive investigations into the fairness issues arising when using LLMs for predictions on tabular data.

## 2.3 IN-CONTEXT LEARNING

Significant improvements for various tasks have been achieved by providing in-context examples to LLMs (Brown et al., 2020; Liu et al., 2022; 2023b). However, previous research by Min et al. (2022); Wei et al. (2023b); Lyu et al. (2023) illustrate that the effective performance of in-context learning largely hinges on semantic priors rather than learning the input-label mapping (Akyürek et al., 2022; Xie et al., 2022; Von Oswald et al., 2023) and the labels of the in-context examples might not play a crucial role in in-context learning, with flipped or random labels sometimes having minimal impact on performance. Despite these findings, the predominant focus of existing investigation of in-context learning remains on conventional natural language processing tasks (Zhao et al., 2021; Min et al., 2022; Wei et al., 2023a;b), largely overlooking the domain of tabular data. Furthermore, the fairness of in-context learning and the impact of flipped labels on this fairness is yet to be thoroughly investigated.

## 3 EXPERIMENTAL SETUP

In this section, we outline the general setup of the experiments conducted in our work.

### 3.1 MODELS

In our work, we focus our experiments on GPT-3.5 (engine `GPT-3.5-turbo`) - an LLM released by OpenAI, trained with instruction tuning (Sanh et al., 2022; Wei et al., 2022) and reinforcement learning from human feedback (RLHF) (Ouyang et al., 2022), aligning LLMs with human preferences. Furthermore, we also compare its performance with conventional machine learning models in order to gain insight into the propagation of biases found within LLMs, which are likely mirrored in traditional models as well, consequently, offering valuable additional perspectives on the biases inherent in the training of LLMs. For this, we employ two widely used models for tabular data i.e, Random Forests (RF) and a shallow Neural Network (NN) of 3 layers. We provide additional implementation details for these two models in the Appendix B.

### 3.2 DATASETS AND PROTECTED ATTRIBUTES

To explore the fairness of LLMs in making predictions for tabular data, we utilize the following three widely used tabular datasets for assessing the fairness of traditional ML models: *Adult Income* (**Adult**) Dataset (Becker & Kohavi, 1996), **German Credit** Dataset (Dua & Graff, 2019), and *Correctional Offender Management Profiling for Alternative Sanctions* (**COMPAS**) Dataset (Larson et al., 2016). In this section, we introduce each dataset and discuss its associated protected attributes.

**Adult**    The *Adult Income* dataset (Adult) is extracted from the 1994 U.S. Census Bureau database. The task is to predict whether a person earns more than $50,000 per year based on their profile data (*greater than 50K* or *less than or equal to 50K*). The original Adult Income Dataset contains 14 features. Following previous work (Slack & Singh, 2023), we retain only 10 features: *"workclass"*, *"hours per week"*, *"sex"*, *"age"*, *"occupation"*, *"capital loss""*, *"education"*, *"capital gain"*, *"marital status"*, and *"relationship"*. Our analysis on Adult primarily focuses on *sex* as the protected attribute, and *female* is acknowledged as a disadvantaged group.

**German Credit**    The German Credit dataset is used to classify individuals based on their profile attributes as good or bad credit risks (*good* or *bad*). The raw dataset comprises 20 attributes. Consistent with previous work, we only retain the following features: *"age"*, *"sex"*, *"job"*, *"housing"*, *"saving accounts"*, *"checking account"*, *"credit amount"*, *"duration"*, and *"purpose"*. Same with Adult, *sex* is considered as a protected attribute in the German Credit dataset and *female* as the marginalized group.

**COMPAS**    The COMPAS dataset comprises the outcomes from the *Correctional Offender Management Profiling for Alternative Sanctions* commercial algorithm, utilized to evaluate a convicted criminal's probability of reoffending. Known for its widespread use by judges and parole officers, COMPAS has gained notoriety for its bias against African-Americans. The raw COMPAS Recidivism dataset contains more than 50 attributes. Following the approach of Larson et al. (2016), we

perform necessary preprocessing, group *"race"* into *African-American* and *Not African-American*, and only consider the features *"sex"*, *"race"*, *"age"*, *"charge degree"*, *"priors count"*, *"risk"* and *"two year recid"* (target). We frame the task as predicting whether an individual will recidivate in two years (*Did Not Reoffend* or *Reoffended*) based on their demographic and criminal history. For the COMPAS dataset, we consider *race* as the protected attribute.

A detailed description for each feature of the considered datasets is provided in Appendix A.

### 3.3 SERIALIZATION AND PROMPT TEMPLATES

To employ the LLM for making predictions on these tabular datasets, each data point is first serialized as text. Following previous works on LLM for tabular predictions (Hegselmann et al., 2023; Slack & Singh, 2023), we format the feature names and values into strings as "$f_1 : x_1, \ldots, f_d : x_d$", and prompt to LLM along with a task description, as illustrated following:

```
You must predict if income exceeds $50K/yr. Answer with one of the
following: greater than 50K | less than or equal to 50K.
Example 1 -
workclass: Private
hours per week: 20
sex: Male
age: 17
occupation: Other-service
capital loss: 0
education: 10th
capital gain: 0
marital status: Never-married
relationship: Own-child
Answer: less than or equal to 50K
...

workclass: Private
hours per week: 40
sex: Female
age: 24
occupation: Sales
capital loss: 0
education: Some-college
capital gain: 0
marital status: Never-married
relationship: Own-child
Answer:
```

Figure 1: Prompt Template for **Adult** Dataset.

The example above is from Adult dataset, where text in blue represents the task description, text in green denotes optional few-shot examples (only used in in-context learning), and text in red indicates the test example. We provide the prompt templates for the other two datasets in Appendix C.

### 3.4 EVALUATION METRICS

To assess fairness in the aforementioned datasets, we examine the disparity between different subgroups of protected attributes using the following common fairness metrics: accuracy, F1 score, statistical parity and equality of opportunity. Here, we briefly explain each evaluation metric.

**Accuracy and F1**  As the most basic metric, assessing accuracy among different subgroups ensures that the model delivers consistent performance across all groups, without undue favor to any particular subgroups. Considering that the evaluated datasets may be imbalanced, especially among different subgroups, the F1 Score computes the harmonic mean of precision and recall, offering a balanced perspective between these two metrics.

**Statistical Parity**  Statistical parity is attained when *positive* decision outcomes (e.g., being predicted as a good credit risk) are independent of the protected attributes. This metric assesses whether different subgroups receive similar treatment from the model. For each subgroup $z_i$ of each protected attribute $Z$, we calculate

$$P(\hat{Y} = 1 | Z = z_i).$$

Then we calculate the Statistical Parity Difference (SPD) of this protected attribute as

$$SPD = P(\hat{Y} = 1 | Z = z_1) - P(\hat{Y} = 1 | Z = z_2),$$

where $z_1$ is the minority group and $z_2$ is the majority.

**Equality of Opportunity**  Equality of opportunity requires that qualified individuals have an equal chance of being correctly classified by the model, regardless of their membership in a protected group. This metric ensures equal *true positive* rates between different subgroups, providing equal opportunities for each subgroup. Similar as statistical parity, for equality of opportunity, we calculate the Equal Opportunity Difference (EOD) as

$$EOD = P(\hat{Y} = 1 | Y = 1, Z = z_1) - P(\hat{Y} = 1 | Y = 1, Z = z_2).$$

Each of these metrics offers a different perspective on fairness. For each subgroup from each protected attribute, we will compute every aforementioned metric. A model demonstrating good fairness should show minimal gaps in these fairness metrics between different subgroups. Considering them together can provide a more comprehensive evaluation of the model's fairness across different subgroups, ensuring that individuals are not unfairly disadvantaged based on their membership in a protected group.

## 4 ZERO-SHOT PROMPTING FOR TABULAR DATA

To explore the fairness of LLMs when making predictions on tabular data, we first conduct experiments in a zero-shot setting. We assess the fairness metrics of the outcomes and examine whether LLMs without any finetuning or few-shot examples would be influenced by social biases and stereotypes for tabular predictions. We run all the experiments 5 times and compute the mean and standard deviation.

In Tables 1-3, we present the evaluation of four fairness metrics, namely accuracy (ACC), F1 score (F1), statistical parity (SP), and equality of opportunity (EoO), for GPT-3.5 (engine `GPT-3.5-turbo`), RF and NN models on the **Adult**, **German Credit** and **COMPAS** datasets, respectively. For the Adult and German Credit datasets, the subgroups *female* and *male* are assessed regarding the protected attribute *sex*, identifying *female* as a disadvantaged group. In the COMPAS dataset, we evaluate *race* as protected attributes, recognizing African American (*AA*) as the disadvantaged group.

It is notable that when utilizing LLMs to make predictions for tabular data directly, without any fine-tuning or in-context learning, a significant fairness metric gap between the protected and non-protected groups is observed for GPT-3.5 (highlighted in red). For instance, the EoO difference between *male* and *female* on the *Adult* dataset reaches 0.483, indicating a substantial disadvantage for the *female* group. Additionally, when compared with traditional methods like RF and NN, the bias in zero-shot predictions made by GPT-3.5 is significantly larger for the Adult dataset. This observation suggests an inherent gender bias in GPT-3.5. For COMPAS dataset, the racial bias in zero-shot setting is comparatively lower than RF and NN but is still effectively high.

Exceptionally, GPT-3.5 is extremely biased for German Credit dataset where it classifies almost everything into *'good credit'* class in the zero-shot setting, thus rendering the difference in SP and EoO for both subgroups to be near 0. The accuracy for each subgroup is near to 50%, performing similar to random guessing. The possible reason might be that the German Credit dataset is too challenging for making tabular predictions with LLMs (especially, the features of German Credit

| | | | | ACC | F1 | SP | EoO |
|---|---|---|---|---|---|---|---|
| GPT-3.5-turbo | Zero-Shot | | $f$ | $0.898_{0.001}$ | $0.711_{0.002}$ | $0.065_{0.001}$ | $0.357_{0.000}$ |
| | | | $m$ | $0.742_{0.002}$ | $0.727_{0.002}$ | $0.464_{0.003}$ | $0.840_{0.004}$ |
| | | | $d$ | $0.157_{0.002}$ | $-0.016_{0.002}$ | $-0.399_{0.003}$ | $-0.483_{0.004}$ |
| | Few-shot | Regular | $f$ | $0.899_{0.002}$ | $0.735_{0.003}$ | $0.082_{0.002}$ | $0.429_{0.000}$ |
| | | | $m$ | $0.781_{0.003}$ | $0.749_{0.002}$ | $0.339_{0.003}$ | $0.700_{0.003}$ |
| | | | $d$ | $0.118_{0.004}$ | $-0.014_{0.004}$ | $-0.257_{0.005}$ ↓ | $-0.271_{0.003}$ ↓ |
| | | Label-flipping | $f$ | $0.682_{0.004}$ | $0.590_{0.003}$ | $0.396_{0.006}$ | $0.800_{0.013}$ |
| | | | $m$ | $0.614_{0.002}$ | $0.605_{0.002}$ | $0.545_{0.001}$ | $0.763_{0.003}$ |
| | | | $d$ | $0.068_{0.004}$ | $-0.015_{0.004}$ | $-0.148_{0.006}$ ✓ | $0.037_{0.014}$ ✓ |
| | Finetuning | Regular | $f$ | $0.915_{0.014}$ | $0.773_{0.036}$ | $0.079_{0.002}$ | $0.476_{0.048}$ |
| | | | $m$ | $0.799_{0.005}$ | $0.754_{0.005}$ | $0.269_{0.036}$ | $0.613_{0.053}$ |
| | | | $d$ | $0.116_{0.009}$ | $0.020_{0.039}$ | $-0.190_{0.035}$ ↓ | $-0.137_{0.098}$ ↓ |
| | | Oversampling | $f$ | $0.913_{0.016}$ | $0.770_{0.042}$ | $0.081_{0.004}$ | $0.476_{0.067}$ |
| | | | $m$ | $0.813_{0.007}$ | $0.780_{0.003}$ | $0.310_{0.038}$ | $0.702_{0.048}$ |
| | | | $d$ | $0.100_{0.013}$ | $-0.010_{0.041}$ | $-0.229_{0.030}$ | $-0.226_{0.077}$ |
| | | Undersampling | $f$ | $0.912_{0.015}$ | $0.770_{0.046}$ | $0.086_{0.006}$ | $0.488_{0.084}$ |
| | | | $m$ | $0.794_{0.006}$ | $0.751_{0.001}$ | $0.285_{0.031}$ | $0.631_{0.044}$ |
| | | | $d$ | $0.118_{0.021}$ | $0.018_{0.046}$ | $-0.200_{0.025}$ | $-0.143_{0.040}$ |
| RF | | Regular | $f$ | $0.914_{0.002}$ | $0.767_{0.006}$ | $0.075_{0.003}$ | $0.457_{0.010}$ |
| | | | $m$ | $0.822_{0.005}$ | $0.783_{0.005}$ | $0.269_{0.004}$ | $0.652_{0.004}$ |
| | | | $d$ | $0.092_{0.004}$ | $-0.015_{0.005}$ | $-0.195_{0.003}$ | $-0.195_{0.012}$ |
| | | Oversampling | $f$ | $0.912_{0.006}$ | $0.770_{0.011}$ | $0.084_{0.005}$ | $0.486_{0.012}$ |
| | | | $m$ | $0.824_{0.002}$ | $0.785_{0.002}$ | $0.270_{0.003}$ | $0.656_{0.006}$ |
| | | | $d$ | $0.087_{0.005}$ | $-0.015_{0.01}$ | $-0.185_{0.004}$ | $-0.170_{0.011}$ |
| | | Undersampling | $f$ | $0.917_{0.004}$ | $0.776_{0.011}$ | $0.075_{0.001}$ | $0.471_{0.018}$ |
| | | | $m$ | $0.814_{0.003}$ | $0.771_{0.004}$ | $0.263_{0.002}$ | $0.627_{0.009}$ |
| | | | $d$ | $0.103_{0.005}$ | $0.005_{0.011}$ | $-0.187_{0.001}$ | $-0.156_{0.018}$ |
| NN | | Regular | $f$ | $0.917_{0.003}$ | $0.778_{0.019}$ | $0.081_{0.016}$ | $0.490_{0.068}$ |
| | | | $m$ | $0.819_{0.006}$ | $0.773_{0.015}$ | $0.250_{0.045}$ | $0.614_{0.079}$ |
| | | | $d$ | $0.098_{0.005}$ | $0.006_{0.009}$ | $-0.169_{0.032}$ | $-0.123_{0.033}$ |
| | | Oversampling | $f$ | $0.916_{0.004}$ | $0.794_{0.013}$ | $0.100_{0.016}$ | $0.562_{0.058}$ |
| | | | $m$ | $0.813_{0.012}$ | $0.774_{0.008}$ | $0.286_{0.044}$ | $0.663_{0.056}$ |
| | | | $d$ | $0.103_{0.011}$ | $0.020_{0.018}$ | $-0.186_{0.030}$ | $-0.102_{0.038}$ |
| | | Undersampling | $f$ | $0.904_{0.005}$ | $0.748_{0.014}$ | $0.084_{0.007}$ | $0.452_{0.030}$ |
| | | | $m$ | $0.813_{0.006}$ | $0.774_{0.005}$ | $0.283_{0.023}$ | $0.659_{0.031}$ |
| | | | $d$ | $0.090_{0.006}$ | $-0.026_{0.014}$ | $-0.199_{0.018}$ | $-0.206_{0.031}$ |

Table 1: **Fairness evaluation for Adult dataset**. This table depicts the evaluation of accuracy (ACC), F1 score (F1), statistical parity (SP), and equality of opportunity (EoO) metrics for the subgroup - *female* ($f$) and *male* ($m$) as well as the difference ($d$) between them. We list the protected group first. The significant fairness disparities are highlighted in red. Both in-context learning and finetuning can lead to bias reduction (indicated by ↓), and label-flipped in-context learning can further minimize bias (indicated by ✓).

are ambiguous and vague). This also suggests that, when using LLM to make predictions on tabular data, a potential description of table feature names is favorable.

These findings demonstrate the tendency of LLMs to rely on social biases and stereotypes inherited from their training corpus when applied to tabular data. This implies that using LLMs for predictions on tabular data may incur significant fairness risks, including the potential to disproportionately disadvantage marginalized communities as well as exacerbate social biases and stereotypes present in society. This is particularly concerning given the widespread application of tabular data in high-stake contexts, further magnifying the potential for harm.

## 5 FEW-SHOT PROMPTING FOR TABULAR DATA

As demonstrated in Section 4, employing LLMs for predictions on tabular data reveals significant social biases in a zero-shot setting. Instead of directly utilizing LLMs for zero-shot tabular predictions, this section explores whether including few-shot examples during prompting will reduce or amplify these biases. To delve deeper into the influence of few-shot examples during in-context

| | | | | ACC | F1 | SP | EoO |
|---|---|---|---|---|---|---|---|
| GPT-3.5-turbo | Zero-Shot | | $f$ | $0.471_{0.011}$ | $0.359_{0.021}$ | $0.980_{0.011}$ | $1.000_{0.000}$ |
| | | | $m$ | $0.556_{0.000}$ | $0.357_{0.000}$ | $0.984_{0.000}$ | $0.972_{0.000}$ |
| | | | $d$ | $-0.084_{0.011}$ | $0.002_{0.021}$ | $-0.004_{0.011}$ | $0.028_{0.000}$ |
| | Few-shot | Regular | $f$ | $0.610_{0.013}$ | $0.593_{0.013}$ | $0.348_{0.027}$ | $0.453_{0.029}$ |
| | | | $m$ | $0.606_{0.007}$ | $0.603_{0.008}$ | $0.337_{0.007}$ | $0.450_{0.012}$ |
| | | | $d$ | $0.003_{0.012}$ | $-0.010_{0.011}$ | $0.011_{0.027}$ | $0.003_{0.026}$ |
| | | Label-flipping | $f$ | $0.614_{0.011}$ | $0.606_{0.012}$ | $0.695_{0.011}$ | $0.842_{0.000}$ |
| | | | $m$ | $0.559_{0.013}$ | $0.538_{0.011}$ | $0.638_{0.013}$ | $0.672_{0.023}$ |
| | | | $d$ | $0.056_{0.021}$ | $0.067_{0.021}$ | $0.057_{0.012}$ | $0.170_{0.023}$ |
| | Finetuning | Regular | $f$ | $0.571_{0.067}$ | $0.567_{0.062}$ | $0.619_{0.101}$ | $0.711_{0.186}$ |
| | | | $m$ | $0.548_{0.011}$ | $0.539_{0.023}$ | $0.532_{0.123}$ | $0.569_{0.098}$ |
| | | | $d$ | $0.024_{0.079}$ | $0.029_{0.085}$ | $0.087_{0.022}$ | $0.141_{0.088}$ |
| | | Oversampling | $f$ | $0.536_{0.017}$ | $0.532_{0.012}$ | $0.607_{0.084}$ | $0.658_{0.112}$ |
| | | | $m$ | $0.532_{0.011}$ | $0.523_{0.020}$ | $0.548_{0.079}$ | $0.569_{0.059}$ |
| | | | $d$ | $0.004_{0.028}$ | $0.009_{0.033}$ | $0.060_{0.006}$ | $0.088_{0.053}$ |
| | | Undersampling | $f$ | $0.548_{0.034}$ | $0.547_{0.033}$ | $0.571_{0.034}$ | $0.632_{0.074}$ |
| | | | $m$ | $0.556_{0.000}$ | $0.555_{0.000}$ | $0.444_{0.000}$ | $0.500_{0.000}$ |
| | | | $d$ | $-0.008_{0.034}$ | $-0.008_{0.033}$ | $0.127_{0.034}$ | $0.132_{0.074}$ |
| RF | | Regular | $f$ | $0.581_{0.024}$ | $0.580_{0.025}$ | $0.519_{0.028}$ | $0.611_{0.054}$ |
| | | | $m$ | $0.600_{0.019}$ | $0.588_{0.020}$ | $0.597_{0.022}$ | $0.672_{0.021}$ |
| | | | $d$ | $-0.019_{0.016}$ | $-0.008_{0.016}$ | $-0.078_{0.044}$ | $-0.062_{0.061}$ |
| | | Oversampling | $f$ | $0.576_{0.018}$ | $0.575_{0.018}$ | $0.505_{0.018}$ | $0.589_{0.021}$ |
| | | | $m$ | $0.568_{0.032}$ | $0.552_{0.034}$ | $0.616_{0.025}$ | $0.661_{0.037}$ |
| | | | $d$ | $0.008_{0.034}$ | $0.023_{0.035}$ | $-0.111_{0.013}$ | $-0.072_{0.041}$ |
| | | Undersampling | $f$ | $0.586_{0.024}$ | $0.585_{0.024}$ | $0.533_{0.024}$ | $0.632_{0.047}$ |
| | | | $m$ | $0.575_{0.031}$ | $0.555_{0.037}$ | $0.635_{0.033}$ | $0.683_{0.022}$ |
| | | | $d$ | $0.011_{0.024}$ | $0.031_{0.031}$ | $-0.102_{0.041}$ | $-0.052_{0.039}$ |
| NN | | Regular | $f$ | $0.533_{0.024}$ | $0.533_{0.024}$ | $0.519_{0.028}$ | $0.558_{0.026}$ |
| | | | $m$ | $0.556_{0.017}$ | $0.544_{0.017}$ | $0.584_{0.012}$ | $0.622_{0.022}$ |
| | | | $d$ | $-0.022_{0.037}$ | $-0.012_{0.036}$ | $-0.065_{0.031}$ | $-0.064_{0.026}$ |
| | | Oversampling | $f$ | $0.548_{0.040}$ | $0.547_{0.040}$ | $0.552_{0.028}$ | $0.611_{0.026}$ |
| | | | $m$ | $0.562_{0.026}$ | $0.547_{0.024}$ | $0.603_{0.048}$ | $0.644_{0.057}$ |
| | | | $d$ | $-0.014_{0.037}$ | $0.000_{0.035}$ | $-0.051_{0.061}$ | $-0.034_{0.065}$ |
| | | Undersampling | $f$ | $0.529_{0.049}$ | $0.524_{0.047}$ | $0.467_{0.051}$ | $0.495_{0.042}$ |
| | | | $m$ | $0.495_{0.025}$ | $0.490_{0.023}$ | $0.524_{0.047}$ | $0.517_{0.054}$ |
| | | | $d$ | $0.033_{0.063}$ | $0.035_{0.059}$ | $-0.057_{0.033}$ | $-0.022_{0.061}$ |

Table 2: **Fairness evaluation for German Credit dataset**. This table depicts the evaluation of accuracy (ACC), F1 score (F1), statistical parity (SP), and equality of opportunity (EoO) metrics for the subgroup - *female* ($f$) and *male* ($m$) as well as the difference ($d$) between them.

learning, we not only consider the regular in-context learning approach as detailed in Section 5.1, but we also experiment by flipping the labels of the few-shot examples to further examine their effect on the biases, as discussed in Section 5.2.

Again, for robustness, each experiment is conducted 5 times, with the mean and standard deviation reported.

## 5.1 REGULAR IN-CONTEXT LEARNING

Previous works have demonstrated that LLMs can learn the input-label mappings in context (Akyürek et al., 2022; Xie et al., 2022; Von Oswald et al., 2023). However, the influence of in-context learning on the fairness has not been thoroughly examined. For in-context learning, the test example and task description, along with a few-shot examples, are provided to the LLMs for generating the final predictions. The few-shot examples are inserted before the test example in the prompt, as outlined in Section 3.3. We set the number of in-context examples as 50. For each dataset, we randomly select the in-context examples from the training set for each test example.

In Tables 1-3, we demonstrate that for two of the evaluated datasets (except for COMPAS), the incorporation of few-shot examples brings about performance improvements. Additionally, we observe that incorporating few-shot examples into prompting reduces the fairness metric gap between

| | | | | ACC | F1 | SP | EoO |
|---|---|---|---|---|---|---|---|
| GPT-3.5-turbo | Zero-Shot | | AA | $0.657_{0.005}$ | $0.656_{0.004}$ | $0.395_{0.001}$ | $0.560_{0.002}$ |
| | | | nAA | $0.663_{0.002}$ | $0.588_{0.003}$ | $0.817_{0.002}$ | $0.893_{0.001}$ |
| | | | d | $-0.006_{0.005}$ | $0.068_{0.006}$ | $-0.423_{0.003}$ | $-0.334_{0.002}$ |
| | Few-shot | Regular | AA | $0.633_{0.002}$ | $0.626_{0.002}$ | $0.362_{0.003}$ | $0.495_{0.004}$ |
| | | | nAA | $0.642_{0.001}$ | $0.623_{0.002}$ | $0.614_{0.002}$ | $0.709_{0.002}$ |
| | | | d | $-0.008_{0.003}$ | $0.003_{0.003}$ | $-0.252_{0.003}$ ↓ | $-0.214_{0.005}$ ↓ |
| | | Label-flipping | AA | $0.482_{0.004}$ | $0.482_{0.004}$ | $0.499_{0.004}$ | $0.481_{0.004}$ |
| | | | nAA | $0.412_{0.003}$ | $0.408_{0.003}$ | $0.471_{0.002}$ | $0.404_{0.003}$ |
| | | | d | $0.070_{0.005}$ | $0.074_{0.005}$ | $0.028_{0.005}$ ✓ | $0.077_{0.007}$ ✓ |
| | Finetuning | Regular | AA | $0.611_{0.016}$ | $0.610_{0.016}$ | $0.464_{0.031}$ | $0.576_{0.034}$ |
| | | | nAA | $0.616_{0.013}$ | $0.586_{0.016}$ | $0.657_{0.032}$ | $0.724_{0.029}$ |
| | | | d | $-0.005_{0.017}$ | $0.024_{0.024}$ | $-0.193_{0.030}$ ↓ | $-0.148_{0.027}$ ↓ |
| | | Oversampling | AA | $0.609_{0.007}$ | $0.608_{0.007}$ | $0.494_{0.071}$ | $0.605_{0.066}$ |
| | | | nAA | $0.625_{0.020}$ | $0.583_{0.024}$ | $0.706_{0.037}$ | $0.771_{0.036}$ |
| | | | d | $-0.016_{0.016}$ | $0.025_{0.018}$ | $-0.212_{0.037}$ | $-0.166_{0.046}$ |
| | | Undersampling | AA | $0.591_{0.010}$ | $0.591_{0.012}$ | $0.513_{0.053}$ | $0.605_{0.047}$ |
| | | | nAA | $0.641_{0.008}$ | $0.612_{0.009}$ | $0.663_{0.035}$ | $0.749_{0.037}$ |
| | | | d | $-0.050_{0.016}$ | $-0.021_{0.022}$ | $-0.150_{0.033}$ | $-0.144_{0.039}$ |
| RF | | Regular | AA | $0.662_{0.004}$ | $0.662_{0.004}$ | $0.496_{0.006}$ | $0.660_{0.007}$ |
| | | | nAA | $0.671_{0.004}$ | $0.617_{0.002}$ | $0.767_{0.008}$ | $0.859_{0.009}$ |
| | | | d | $-0.009_{0.007}$ | $0.045_{0.005}$ | $-0.271_{0.011}$ | $-0.199_{0.014}$ |
| | | Oversampling | AA | $0.660_{0.005}$ | $0.660_{0.005}$ | $0.493_{0.010}$ | $0.655_{0.013}$ |
| | | | nAA | $0.671_{0.002}$ | $0.624_{0.002}$ | $0.743_{0.003}$ | $0.839_{0.004}$ |
| | | | d | $-0.010_{0.006}$ | $0.037_{0.006}$ | $-0.250_{0.012}$ | $-0.184_{0.016}$ |
| | | Undersampling | AA | $0.648_{0.002}$ | $0.647_{0.002}$ | $0.491_{0.004}$ | $0.639_{0.004}$ |
| | | | nAA | $0.667_{0.005}$ | $0.614_{0.007}$ | $0.761_{0.006}$ | $0.851_{0.006}$ |
| | | | d | $-0.020_{0.007}$ | $0.033_{0.008}$ | $-0.270_{0.009}$ | $-0.211_{0.008}$ |
| NN | | Regular | AA | $0.666_{0.003}$ | $0.665_{0.002}$ | $0.462_{0.034}$ | $0.630_{0.034}$ |
| | | | nAA | $0.662_{0.003}$ | $0.613_{0.006}$ | $0.742_{0.019}$ | $0.831_{0.017}$ |
| | | | d | $0.005_{0.006}$ | $0.052_{0.007}$ | $-0.280_{0.019}$ | $-0.201_{0.018}$ |
| | | Oversampling | AA | $0.656_{0.001}$ | $0.653_{0.012}$ | $0.507_{0.090}$ | $0.665_{0.101}$ |
| | | | nAA | $0.643_{0.013}$ | $0.580_{0.034}$ | $0.757_{0.107}$ | $0.828_{0.091}$ |
| | | | d | $0.013_{0.014}$ | $0.073_{0.043}$ | $-0.249_{0.049}$ | $-0.163_{0.046}$ |
| | | Undersampling | AA | $0.660_{0.019}$ | $0.657_{0.023}$ | $0.477_{0.078}$ | $0.638_{0.097}$ |
| | | | nAA | $0.657_{0.013}$ | $0.602_{0.026}$ | $0.757_{0.051}$ | $0.839_{0.040}$ |
| | | | d | $0.003_{0.024}$ | $0.055_{0.043}$ | $-0.280_{0.041}$ | $-0.202_{0.064}$ |

Table 3: **Fairness evaluation for COMPAS dataset** for the subgroup - *African American (AA)*, and *Non African American* (*nAA*) as well as the difference (*d*). The significant fairness disparities are highlighted in red. Both in-context learning and finetuning can lead to bias reduction (indicated by ↓), and label-flipped in-context learning can further minimize bias (indicated by ✓).

different subgroups. However, a significant fairness issue still persists. Moreover, for the Adult and COMPAS datasets, the disparity in fairness metrics of in-context learning is more notable when compared to traditional models, such as RF and NN. This highlights the inherent biases embedded within LLMs, which are not solely derived from the task datasets.

## 5.2 LABEL FLIPPING

To delve deeper into the sources of biases within LLMs, we further examine the impact of the labels of in-context examples on fairness. As depicted in Tables 1-3, label flipping significantly reduces biases across all evaluated datasets. And for all evaluated datasets, the difference in statistical parity (SP) and equality of opportunity (EoO) is minimized with label-flipped in-context learning. For example, the absolute gap of EoO on the Adult dataset decreases from 0.483 in zero-shot prompting to 0.037, almost completely eliminating the bias. These findings further corroborates the existence of inherent biases in LLMs.

However, flipped labels lead to a significant drop in predictive performance. Though previous research suggests that the effectiveness of in-context learning predominantly stems from semantic priors, rather than learning the input-label mappings (Min et al., 2022; Wei et al., 2023b) and demon-

strate that the performance of in-context learning is barely affected even with flipped or random labels for in-context examples, the focus of these works lies mainly on traditional natural language processing tasks.

In contrast, we observe that the labels of in-context examples hold substantial influence over predictive performance in our unique setup, where LLMs are deployed for predictions on tabular data. This could be attributed to the limited exposure of these models to tabular data during pre-training, thereby amplifying the role of input-label mapping of in-context examples.

## 6 FINETUNING FOR TABULAR DATA

### 6.1 REGULAR FINETUNING

Finally, we extend our investigation to assess if finetuning the models on the entire training set could aid in diminishing the social biases in LLMs. For GPT-3.5, fine-tuning is executed using the publicly released API from OpenAI. For RF and NN, we provide the training details in Appendix B. We still run all the experiments 5 times and compute the mean and standard deviation.

In Tables 1-3, we show that finetuning effectively reduces unfairness in all datasets, making them comparable and sometimes significantly better in terms of SP and EoO when compared to RF and NN. For example, the absolute difference in EoO after finetuning on Adult dataset is 0.0714, which is lower than 0.123 difference of a NN.

### 6.2 RESAMPLING

We further explore the potential of resampling, a method frequently employed to enhance fairness in machine learning model training, particularly in scenarios where there is a significant class imbalance or bias in the data. To this end, we evaluate two approaches: oversampling the minority group and undersampling the majority group. As depicted in Tables 1-3, resampling fails to mitigate the social biases in LLMs when making tabular predictions, even though we demonstrate that oversampling generally reduces social biases for both RF and NN, except for a few instances such as, oversampling in NN for adult dataset worsens the fairness.

Our finetuning experiments show that the social biases inherited from LLM's pre-training data which are evident when making predictions on tabular data, can sometimes be mitigated through finetuning. Nevertheless, unlike the consistent outcomes typically seen in traditional machine learning models, like RF and NN, data resampling does not consistently produce similar results for finetuning LLMs.

## 7 CONCLUSION

In this work, we thoroughly investigate the under-explored problem of fairness of large language models (LLMs) for tabular tasks. Our study unfolds in several phases. Initially, we assess the inherent fairness displayed by LLMs, comparing their performance in zero-shot learning scenarios against traditional machine learning models like random forests (RF) and shallow neural networks (NN). Furthermore, we investigate how LLMs learn and propagate social biases when subjected to few-shot in-context learning, label-flipped in-context learning, fine-tuning, and data resampling techniques.

Our discoveries shed light on several key insights. We find that LLMs tend to heavily rely on the social biases inherited from their pre-training data when making predictions, which is a concerning issue. Moreover, we observe that few-shot in-context learning can partially mitigate the inherent biases in LLMs, yet it cannot entirely eliminate them. A significant fairness metric gap between different subgroups persists, and exceeds that observed in RF and NN. This observation underscores the existence of biases within the LLMs themselves, beyond just the task datasets. Additionally, label-flipping applied to the few-shot examples effectively reverses the effects of bias, again corroborating the existence of inherent biases in LLMs. However, as expected, this leads to a loss in predictive performance. Besides, our work reveals that while fine-tuning can sometimes improve the fairness of LLMs, data resampling does not consistently yield the same results, unlike what is typically observed in traditional machine learning models.

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

## A  DESCRIPTION FOR EACH FEATURE IN EACH DATASET

We provide a detailed description of each feature from the datasets evaluated in our paper.

### A.1  ADULT

The original Adult Income Dataset contains 14 features an the target *Income*, as described in Table 4. Following prior work (Slack & Singh, 2023), we omit *Education-Num* and *Fnlwgt* as they are not crucial for income prediction, along with *Race* and *Native-Country*, to center our attention on *Sex* as the protected attribute.

| Feature | Type | Description |
|---|---|---|
| Age | Continuous | Represents the age of an individual. |
| Workclass | Categorical | Indicates the type of employment, such as private, self-employed, or government. |
| *Fnlwgt* | Continuous | Stands for "final weight" and is a numerical value used in sampling for survey data. |
| Education | Categorical | Specifies the highest level of education attained by the individual, such as high school, bachelor's degree, etc. |
| *Education-Num* | Continuous | Represents the numerical equivalent of the education level. |
| Marital-Status | Categorical | Describes the marital status of the individual, including categories like married, divorced, or single. |
| Occupation | Categorical | Indicates the occupation of the individual, such as managerial, technical, or clerical work. |
| Relationship | Categorical | Specifies the individual's role in the family, such as husband, wife, or child. |
| Race | Categorical | Represents the individual's race or ethnic background. |
| Sex | Categorical | Indicates the gender of the individual, either male or female. |
| Capital-Gain | Continuous | Refers to the capital gains, which are profits from the sale of assets, of the individual. |
| Capital-Loss | Continuous | Represents the capital losses, which are losses from the sale of assets, of the individual. |
| Hours-Per-Week | Continuous | Denotes the number of hours worked per week by the individual. |
| *Native-Country* | Categorical | Specifies the native country or place of origin of the individual. |
| Income (target) | Binary | The target variable indicating whether an individual's income exceeds a certain threshold, typically $50,000 per year. |

Table 4: Features in the original **Adult** dataset. Those not used in our work are shown in *italics*.

### A.2  GERMAN CREDIT

The original German Credit Dataset contains 20 features, as detailed in Table 5. For simplicity and consistency with prior work, only the features not shown in *italics* are retained in our work. Furthermore, we extract *Sex* as an additional protected attribute from the *Personal Status and Sex* feature.

### A.3  COMPAS

The raw COMPAS Recidivism dataset contains more than 50 attributes. Following the approach of Larson et al. (2016), we carry out the necessary preprocessing. More specifically, we group the *race* attribute into *African-American* and *Not African-American*, and consider only the features *sex*, *race*, *age*, *charge degree*, *priors count*, *risk*, and *two-year recid* (target). We frame the task as predicting

| Feature | Type | Description |
|---|---|---|
| Credit Amount | Continuous | The amount of credit requested by the applicant. |
| Duration | Continuous | The duration of the credit in months. |
| *Installment Rate* | Ordinal | The installment rate in percentage of disposable income. |
| *Residence Since* | Ordinal | The number of years the applicant has lived at their current residence. |
| Age | Continuous | The age of the applicant. |
| *Number of Existing Credits* | Ordinal | The number of existing credits at this bank. |
| *Number of Dependents* | Ordinal | The number of dependents of the applicant. |
| Checking Account Status | Categorical | The status of the applicant's checking account, such as "no checking, "<0 DM," "0-200 DM," or "no known checking." |
| *Credit History* | Categorical | The credit history of the applicant, including categories like "critical/other existing credit," "existing paid," "delayed previously," etc. |
| Purpose | Categorical | The purpose of the credit, such as "radio/tv," "education," "new car," etc. |
| Savings Account | Categorical | The status of the applicant's savings account/bonds, including categories like "unknown/none," "<100 DM," "500-1000 DM," etc. |
| *Employment Since* | Categorical | The duration of the applicant's current employment, such as "unemployed," "<1 year," "4-7 years," etc. |
| *Personal Status and Sex* | Categorical | The personal status and sex of the applicant, including categories like "male single," "female div/dep/mar," etc. |
| *Other Debtors/Guarantors* | Categorical | Indicates the presence of other debtors/guarantors, such as "none," "guarantor," "co applicant." |
| *Property* | Categorical | Describes the type of property owned by the applicant, such as "real estate," "life insurance," "car or other," etc. |
| *Other Installment Plans* | Categorical | The presence of other installment plans. |
| Housing | Categorical | The housing situation of the applicant, such as "own," "for free," "rent." |
| Job | Categorical | The type of job held by the applicant, including categories like "skilled," "unskilled resident," "high qualif/self emp/mgmt," etc. |
| *Telephone* | Binary | Indicates whether the applicant has a telephone (yes/no). |
| *Foreign Worker* | Binary | Indicates whether the applicant is a foreign worker (yes/no). |
| Risk (target) | Binary | The target variable indicating credit risk (good/bad). |

Table 5: Features in the original **German Credit** dataset. Those not used in our work are shown in *italics*. Additionally, from the original feature *Personal Status and Sex*, we extract *Sex* as a protected attribute.

whether an individual will recidivate within two years (*Did Not Reoffend* or *Reoffended*), based on their demographic and criminal history. Due to page limitations, we provide descriptions for only the features used in our work in Table 6.

## B   RF AND NN HYPERPARAMETERS

For RF, we fix number of trees to 100 for all datasets as well as models. For NN, we use a 3 hidden-layered network with hyperparameters described in Table 7.

| Feature | Type | Description |
|---|---|---|
| Sex | Categorical | The gender of the individual. |
| Race | Categorical | The race of the individual, grouped into *African-American* and *Not African-American*. |
| Age | Continuous | The age of the individual. |
| Charge Degree | Categorical | The degree of the charge against the individual. |
| Priors Count | Continuous | The number of prior convictions or charges. |
| Risk | Categorical | The risk assessment for recidivism. |
| Two-Year Recid (target) | Binary | The target variable indicating whether an individual recidivated within two years. |

Table 6: Features in the **COMPAS** Recidivism Dataset (Preprocessed).

| | h1 | h2 | h3 | lr | batch size | epochs |
|---|---|---|---|---|---|---|
| **Adult** | 16 | 64 | 16 | 0.07 | 128 | 300 |
| **German Credit** | 64 | 64 | 32 | 0.07 | 128 | 300 |
| **COMPAS** | 64 | 128 | 64 | 0.09 | 128 | 300 |

Table 7: Hyperparameters for all datasets for a 3 layer neural network, where h1, h2 and h3 represent the number of neurons in first, second and third hidden layers respectively, lr represents the learning rate, and is followed by the batch size and number of epochs the models are trained for.

## C    PROMPT TEMPLATES FOR EACH DATASET

Beyond the Adult dataset, we provide the serialization and prompt templates utilized in our work for the German Credit and COMPAS datasets here.

### C.1    GERMAN CREDIT

```
Predict the credit risk of a German bank customer based on their profile
data. Answer with one of the following: bad | good.
Example 1 -
Age: 25 - 45
Sex: female
Job: highly skilled
Housing: rent
Saving accounts: little
Checking account: little
Credit amount: 2606
Duration: 21
Purpose: radio/TV
Answer: good

Age: 25 - 45
Sex: male
Job: skilled
Housing: own
Saving accounts: little
Checking account: little
Credit amount: 1345
Duration: 18
Purpose: radio/TV
Answer:
```

Figure 2: Prompt Template for **German Credit** Dataset.

### C.2    COMPAS

```
Predict whether an individual will recidivate with in two years based on
demographic and criminal history. Answer with one of the following: Did
Not Reoffend | Reoffended.
Example 1 -
sex: Male
race: African-American
age cat: 25 - 45
c charge degree: F
priors count: 0
risk: Low
Answer: Did Not Reoffend

sex: Male
race: African-American
age cat: 25 - 45
c charge degree: M
priors count: 13
risk: High
Answer:
```

Figure 3: Prompt Template for **COMPAS** Dataset.

