# OpenReview forum: "Investigating the Fairness of Large Language Models for Predictions on Tabular Data"
_ICLR.cc/2024/Conference — ICLR 2024 Conference Withdrawn Submission_

### Official Review · Reviewer_qNdb · 2023-10-29

**Soundness:** 2 fair
**Presentation:** 3 good
**Contribution:** 2 fair
**Rating:** 3
**Confidence:** 4

**Summary:**

This paper investigates the fairness of using Large Language Models (LLMs) for predictions on tabular data. It reveals that LLMs inherit and exhibit social biases, emphasizing the inherent nature of these biases and the associated fairness risks when applying LLMs to tabular tasks. The study also shows that bias mitigation techniques have a moderate effect, and the fairness gap remains larger in LLMs compared to traditional machine learning models, highlighting the presence of inherent bias within LLMs.

**Strengths:**

- The paper provides a thorough examination of the use of Large Language Models (LLMs) in tabular data prediction tasks.
- The paper backs its claims with a series of empirical experiments, demonstrating that LLMs tend to inherit social biases from their training data and illustrating the impact on fairness in tabular predictions.
- The study goes beyond identifying biases and delves into bias mitigation techniques. The observation that in-context learning and fine-tuning have a moderate effect on reducing biases offers practical insights for improving the fairness of LLMs.

**Weaknesses:**

- The paper may lack novelty in its findings and insights, as it reinforces the existing understanding that Large Language Models (LLMs) exhibit biases. While it provides empirical evidence in the context of tabular data, it doesn't substantially advance the field's understanding of LLM biases.
- The paper may not introduce groundbreaking or exciting concepts, which can make it less engaging for readers. Research in LLMs and fairness requires innovative approaches to capture attention and stand out.
- The paper utilizes API-based ChatGPT, which may suffer from potential data contamination that the datasets used for training LLMs might be included in their training process. Besides, it's possible that the training data of ChatGPT contains much more information related to the used datasets, which makes the comparison between it and RF/NN unfair. The paper doesn't provide concrete evidence of this issue, which can weaken the argument and leave room for ambiguity.
- It is not clear about the choices of hyperparameters in RF/NN as well as ChatGPT fine-tuning. The appendix only mentioned the final choices of parameters used in RF/NN but there is no illustration about the reasons.
- While the paper compares LLMs with traditional machine learning models like Random Forest and Neural Networks, it may not explore a broader range of machine learning approaches or propose novel techniques for mitigating biases in LLMs.
- The paper doesn't provide substantial guidance or recommendations for addressing the fairness challenges LLMs pose in practice. Readers may benefit from more practical, ethical, and policy-oriented insights.

**Questions:**

Please refer to Section "Weaknesses".

---

### Official Review · Reviewer_eUu3 · 2023-11-01

**Soundness:** 2 fair
**Presentation:** 3 good
**Contribution:** 2 fair
**Rating:** 3
**Confidence:** 4

**Summary:**

The paper evaluates the fairness of LLMs (GPT-3.5-turbo) on tabular data tasks (Adult, German Credit and COMPAS) compared to Random Forest and shallow Neural Network baselines. They experiment with zero-shot and few-shot prompting as well as model finetuning, concluding that the fairness metric gap between different subgroups is still larger than that in traditional machine learning models.

**Strengths:**

* It is an important and timely topic.
* The textual part of the paper is easy to read, follow and understand.
* Preliminary experiments evaluating the fairness of GPT-3.5-turbo on tabular data is conducted.

**Weaknesses:**

Unfortunately, I believe that the technical contributions and the experimental setup are very limited.

* Only one LLM model is evaluated (GPT-3.5-turbo)
* Only one prompting template is tested
* I am not convinced that the selected traditional machine learning baselines are strong enough. For example, there is a plethora of bias mitigation techniques presented in the literature and evaluated on the same datasets: adversarial learning, fair representation learning, post-processing techniques, etc. Moreover, the accuracies of baseline models (NN & RF) on the German dataset seem unusually low. I believe, it should be around 75%.
* I find the analysis of the results to be relatively shallow.
* The tables with the results (Table 1-3) are quite overwhelming. It is difficult to look and interpret them. Is it possible to present the results more effectively (e.g., in the form of plots or graphs)?

Minor: I think the fairness metrics (SPD and EOD) are usually presented as absolute values.

**Questions:**

In addition to the above comments, I have the following questions:

Q1: What is the unfairness exhibited by other popular LLMs? What is the impact of the size of the models on the performance and the unfairness?

Q2: What is the impact of the different prompting formats? You mention that "the features of German Credit are ambiguous and vague ...  when using LLM to make predictions on tabular data, a potential description of table feature names is favorable". Did you experiment with this? For example, Hegselmann et al. (2023) consider manual templates, table-to-text and LLMs to construct the input prompt.

Q3: You mention that a goal of the paper is to "discern the underlying information sources upon which LLMs rely when making tabular predictions". However, it is not clear what biases exactly the LLM model is referring to when making the predictions. It is not clear why LLM finetuning reduces unfairness. Is it simply because it gets more accurate or "unlearns"/reduces some of the biases? If so, the mechanism through which this happens is not clear.

Q4: Did you try any chain-of-thought prompting or other strategies which have been found helpful when solving tasks with LLM?

Minor: the study does not take into account potential issues with generalization over time. E.g., the reference of the Adult dataset is from 1996, but I would expect that current LLMs might have a more "modern" and contemporary view.

---

### Official Review · Reviewer_rhXH · 2023-11-01

**Soundness:** 1 poor
**Presentation:** 2 fair
**Contribution:** 3 good
**Rating:** 3
**Confidence:** 4

**Summary:**

The authors investigate three measures of bias in GPT-3.5-turbo on tabular data tasks in zero and few-shot settings, and find more bias in language models than in traditional machine learning systems.

**Strengths:**

The problem being tackled is of societal importance, especially as language models are being used in an increasing amount of tasks. The datasets used address very important and high-stakes applications, including recidivism and credit scoring. They also identify multiple measures of fairness that are important to consider in these applications.

**Weaknesses:**

The paper could benefit from:
1) First optimizing for improvement of task performance before making claims on fairness. From a scientific standpoint, this will help with distinguishing between lack of task understanding appearing in fairness metrics as disparity, versus bias. From an applications standpoint, high performance is a pre-requisite (alongside bias measures) for usage, thus a more comprehensive study on models with high performance will be more impactful to both audiences.
2) Experiments on more language models aside from GPT-3.5-turbo.

**Questions:**

What hypothesis do you have to why resampling fails to mitigate social biases in LLMs?

---

### Official Review · Reviewer_Vp7m · 2023-11-01

**Soundness:** 2 fair
**Presentation:** 2 fair
**Contribution:** 2 fair
**Rating:** 5
**Confidence:** 4

**Summary:**

This work proposes to study the fairness concerns of LLM-based approaches towards reasoning on tabular data. Experiments on three datasets demonstrate that LLMs do exhibit biases towards different demographics when doing tasks with tabular data, and such biases could be mitigated by label flipping, fine-tuning, and more, to varying extents.

**Strengths:**

- LLM fairness is an important research question
- the combination of fairness research and tabular reasoning is interesting

**Weaknesses:**

- Is there support from previous works or fairness/bias theory for the two metrics, statistical parity and equality of opportunity? There is no reference included in those sections as of now.

- Tables 1, 2, and 3 feel repetitive: one ocean of numbers after another, with no clear highlights or insights included in the table. I would suggest finding a way to synthesize them and present them in a concise way.

- This is more of a discussion point: In Tables 1-2, since the authors highlighted the performance for women (f), men (m), and their difference (d), I assume that the objective here is to let d be as little as possible. Empirically, d will never be 0, so what should be a reasonable goal/objective when we are evaluating things in this way? In addition, while the datasets may only contain male and female labels, it might be helpful to acknowledge non-binary people and discuss how this work could be better informed in this way, perhaps in an ethical considerations section.

- While the authors acknowledge that there is much research on LLM for tabular data, the approach investigated in this work is only plain prompting. I wonder if there might be more advanced approaches for LLM tabular reasoning and if they could be included in the studies. This will impact the generalizability and impact of the findings.

- A minor point, but it might be nice to also have an open-source model in the experiments, at least a subset of it. gpt-3.5-turbo goes through periodic updates, while an open-source model might provide better reproducibility. Not necessarily a 70b llama2, but anything on the 7b scale should suffice.

- One major concern is that the experiments/analyses in this work are a bit underwhelming in their current shape. Three huge and monotonous tables are presented in the main paper, without any further analysis, particularly on the qualitative side. Decisions could be made to streamline the main paper content and include additional results that dig deeper into those fairness issues.

- Figure 1 might be taking too much valuable space for additional analysis and insights: prompt templates are generally kept to the appendix and I suggest the authors do the same.

- Some more up-to-date works on LLM fairness could be included to motivate this work better, why biases are picked up from pretraining data and propagated to downstream tasks, and more. Some pointers include [1-2], while I encourage the authors to potentially expand the related work discussion and better position this work.

[1] Omar Shaikh, Hongxin Zhang, William Held, Michael Bernstein, and Diyi Yang. 2023. On Second Thought, Let’s Not Think Step by Step! Bias and Toxicity in Zero-Shot Reasoning. In Proceedings of the 61st Annual Meeting of the Association for Computational Linguistics (Volume 1: Long Papers), pages 4454–4470, Toronto, Canada. Association for Computational Linguistics.

[2] Shangbin Feng, Chan Young Park, Yuhan Liu, and Yulia Tsvetkov. 2023. From Pretraining Data to Language Models to Downstream Tasks: Tracking the Trails of Political Biases Leading to Unfair NLP Models. In Proceedings of the 61st Annual Meeting of the Association for Computational Linguistics (Volume 1: Long Papers), pages 11737–11762, Toronto, Canada. Association for Computational Linguistics.

**Questions:**

please see above

---

### Official Review · Reviewer_fjE3 · 2023-11-05

**Soundness:** 2 fair
**Presentation:** 3 good
**Contribution:** 3 good
**Rating:** 5
**Confidence:** 4

**Summary:**

This paper conduct a series of experiments on several datasets, which show that LLMs tend to inherit social biases from their training data which significantly impact their fairness in tabular prediction tasks.

**Strengths:**

1、The research point that the paper focuses on is very worthy of research and has great practical significance.
2、The paper proves the problem of social bias in the LLM through experiments from different perspectives, and provides sufficient experiments to discuss the problem.
3、The paper verifies the relationship between few-shot in context learning, label-applied in context learning, fine tuning, and data sampling techniques and social bias. The analyses from multiple perspectives provide the reference for the feasibility of methods for alleviating bias in LLMs.

**Weaknesses:**

1、The entry point of the paper is the issue of social bias in the LLMs, but the experiment was only validated on GPT-3.5 and not on other large models. Using one result of LLM to represent all LLMs is a little of unfair.
2、The paper did not discuss whether there is social bias in the selected datasets, and only speculated whether the social bias in the LLM is not sufficient based on the effectiveness of the RF and NN models. After all, we cannot know whether the RF and NN models have learned this bias.
3、The paper lacks a certain explanation for some conflicting results.

**Questions:**

1、The paper was only validated on GPT-3.5 and cannot summarize all LLMs. It is recommended that the authors can conduct experiments on other LLMs.
2、Suggest the author to analyze whether the selected datasets have social bias. If there is no social bias in the dataset, it can prove the validity of the experimental results. Otherwise, it should be explained why the experimental results infer the credibility of social bias in LLMs under biased datasets.
3、Some of the results of the paper lack a deeper explanation, such as label flipping, which greatly reduces the accuracy of prediction while reduces the difference in EoO. Can we believe that label flipping disrupts the model's cognition, which can cause LLM unable to infer labels based on inherited knowledge? So, is it not a very reasonable approach to alleviate social bias through label flipping while reducing the reliability and helpfulness of the LLM?

---

### Official Review · Reviewer_88s7 · 2023-11-10

**Soundness:** 2 fair
**Presentation:** 3 good
**Contribution:** 2 fair
**Rating:** 5
**Confidence:** 4

**Summary:**

In light of the significant advancements in large language models (LLMs) across various domains, the issue of inherent bias in these models has emerged as a critical area of research. Despite ongoing efforts to address this, the specific application of LLMs to tabular data – crucial in many high-stakes domains – has received less attention. This study conducts a series of experiments to examine the bias in LLMs, particularly concerning their application to tabular datasets. Central to this investigation is GPT-3.5, benchmarked against traditional machine learning models like Random Forests and a basic neural network. The evaluation utilizes three common benchmarks, incorporating key protected attributes like sex and race, to measure fairness. The findings reveal that GPT-3.5 often carries over biases from their pre-training data, affecting fairness in tabular predictions, notably in zero-shot contexts. Techniques like in-context learning demonstrated some potential in mitigating bias, yet they do not fully eliminate it compared to conventional baseline models. Interestingly, while fine-tuning can enhance fairness, the study finds that data resampling – a prevalent method in traditional models – is less effective with LLMs. The paper emphasizes the necessity for more advanced approaches to effectively eliminate biases in LLMs, particularly for tabular data.

**Strengths:**

1. The paper focuses on the under-explored issue of bias in LLMs, specifically in the context of tabular data predictions, a fresh angle in fairness research.

2. The comparison of GPT-3.5 against traditional models, using methods such as zero-shot, few-shot, and fine-tuning, provides a well-designed combination of experiments and a well-rounded analysis. This is also a good reference for future work.

3. The paper is well-organized, with a clear presentation, enhancing readability and understanding.

**Weaknesses:**

1. The empirical contributions are appreciated; however, the insights offered by this work may not be substantial. The focus on tabular data is interesting, yet it is a widely held expectation, potentially supported by existing research, that LLMs could carry biases from pre-training into fine-tuning phases.
2. While certain findings are interesting, they require further exploration to yield deeper insights.

**Questions:**

1. The experimental results indicate that some bias mitigation techniques, such as resampling, only work for some cases. Is there a detailed analysis or explanation for this?
2. While the paper refers to LLMs in general, the empirical evaluation is limited to GPT-3.5. Are there experimental results involving other LLMs to support the broader claims? If so, are these results consistent with those observed for GPT-3.5?